# Sirt6-Mediated Cell Death Associated with Sirt1 Suppression in Gastric Cancer

**DOI:** 10.3390/cancers16020387

**Published:** 2024-01-16

**Authors:** Ji Hyun Seo, Somi Ryu, So Young Cheon, Seong-Jun Lee, Seong-Jun Won, Chae Dong Yim, Hyun-Jin Lee, Young-Sool Hah, Jung Je Park

**Affiliations:** 1Department of Pediatrics, Institute of Health Science, College of Medicine, Gyeongsang National University, Jinju 52725, Republic of Korea; seozee@hanmail.net; 2Institute of Medical Science, Gyeongsang National University, Jinju 52725, Republic of Korea; rsm486@hanmail.net (S.R.); imgun007@hanmail.net (C.D.Y.); 3Department of Otorhinolaryngology-Head and Neck Surgery, College of Medicine, Gyeongsang National University, Gyeongsang National University Hospital, Jinju 52727, Republic of Korea; 4Biomedical Research Institute, Gyeongsang National University Hospital, Jinju 52727, Republic of Korea; sy7231@daum.net; 5Department of Convergence of Medical Sciences, Gyeongsang National University, Jinju 52725, Republic of Korea; 6Department of Otorhinolaryngology-Head and Neck Surgery, College of Medicine, Chung-Ang University, Chung-Ang University Gwangmyeong Hospital, Gwangmyeong 06973, Republic of Korea

**Keywords:** sirtuins, Sirtuin1, Sirtuin6, gastric cancer

## Abstract

**Simple Summary:**

This study highlights the potential anti-cancer effects of the sirtuin (Sirt) family in gastric cancer. Among the seven mammalian sirtuin proteins, Sirt6 and Sirt1 seem to be key factors of gastric cancer cell death. To be more specific, Sirt6 overexpression in association with Sirt1 suppression induces reactive oxygen species (ROS) and mouse double minute 2 homolog (MDM2) expression, which result in cancer cell death and a reduction in tumor volume. These findings, by supporting Sirt6- and Sirt1-mediated gastric cancer cell death via ROS regulation, suggest new potential therapeutic targets for the future treatment of gastric cancer.

**Abstract:**

Background: Gastric cancer, one of the leading causes of cancer-related death, is strongly associated with *H. pylori* infection, although other risk factors have been identified. The sirtuin (Sirt) family is involved in the tumorigenesis of gastric cancer, and sirtuins can have pro- or anti-tumorigenic effects. Methods: After determining the overall survival rate of gastric cancer patients with or without Sirt6 expression, the effect of Sirt6 upregulation was also tested using a xenograft mouse model. The regulation of Sirt6 and Sirt1, leading to the induction of mouse double minute 2 homolog (MDM2) and reactive oxygen species (ROS), was mainly analyzed using Western blotting and immunofluorescence staining, and gastric cancer cell (SNU-638) death associated with these proteins was measured using flow cytometric analysis. Results: Sirt6 overexpression led to Sirt1 suppression in gastric cancer cells, resulting in a higher level of gastric cancer cell death in vitro and a reduced tumor volume. ROS and MDM2 expression levels were upregulated by Sirt6 overexpression and/or Sirt1 suppression according to Western blot analysis. The upregulated ROS ultimately led to gastric cancer cell death as determined via Western blot and flow cytometric analysis. Conclusion: We found that the upregulation of Sirt6 suppressed Sirt1, and Sirt6- and Sirt1-induced gastric cancer cell death was mediated by ROS production. These findings highlight the potential of Sirt6 and Sirt1 as therapeutic targets for treating gastric cancer.

## 1. Introduction

Sirtuins are nicotinamide adenine dinucleotide (NAD+)-dependent signaling proteins that regulate various metabolic processes such as energy metabolism, inflammation, and apoptosis in a wide range of organisms ranging from yeast to mammals [1]. They play essential roles in aging-related pathological conditions including diabetes, cardiovascular diseases, neurodegenerative diseases, and cancer [2]. Sirtuins play a dual role in cancer, and some sirtuins suppress tumorigenesis whereas others promote it [3,4]. Sirtuins have anti-tumorigenic activity because they inhibit senescence and allow for unchecked cell division [4,5]. However, sirtuin family members can protect DNA from unfavorable environmental conditions such as oxidative stress, maintain genomic stability, and regulate replicative lifespan via acting as pro-tumorigenic proteins [4,5]. Although the roles of sirtuins in tumorigenesis have been studied extensively, the underlying mechanisms are complex [3,4].

Among the seven mammalian sirtuin family proteins (Sirt1-Sirt7), Sirt6 is the most studied tumor suppressor protein [3]. The anti-tumorigenic activity of Sirt6 is mediated by anti-apoptotic factors influencing cancer cell survival and development [6,7]. It seems to be regulated by various molecular mechanisms including Forkhead box protein O1 (FOXO1) and the YAP signaling pathway, the targeting of p53, and the inhibition of the phosphatidylinositol 3-kinase (PI3K)/AKT pathway [3,4]. Sirt1 has a dual function as a pro-tumorigenic and anti-tumorigenic factor depending on the environment and cell type [8,9,10]. Sirt1 is overexpressed in many solid tumors, and its dual effect has been demonstrated, especially in gastric cancer (Table 1). Sirt1 influences cell viability and the apoptosis of certain solid cancer cells by regulating tumor-growth-associated signaling pathways such as the Wnt signaling and JAK2/STAT3 pathways [9,10].

Interestingly, the anti-cancer effect of Sirt6 seems more apparent in interaction with Sirt1, inducing MDM2-dependent Sirt1 degradation or potentiating the DNA damage response [6,7]. Although the anti-cancer effects of Sirt6, especially concerning the inhibition of Sirt1, have been documented for various cancers, this phenomenon remains unexplored in the context of gastric cancer [6]. Sirt6 and Sirt1 are involved in signaling networks such as those mediated by the Forkhead-box transcription factor family, nuclear factor kappa B (NF-κB), and tumor suppressor p53, as well as in the generation of ROS [6]. ROS production is increased by Sirt6-mediated Sirt1 suppression through the regulation of MDM2, and the relationship between Sirt6 and Sirt1 under oxidative stress conditions is essential for carcinogenesis [40]. In this study, we examined the anti-tumorigenic effect of Sirt6 on inducing ROS-related gastric cancer cell death by suppressing Sirt1 expression.

## 2. Materials and Methods

### 2.1. Patient Samples and Data Collection

This study was approved by the Institutional Review Board of Gyeongsang National University Hospital. Tissue samples were obtained from patients with stage I–IV gastric cancer who had been diagnosed at Gyeongsang National University Hospital between 2001 and 2010. For immunohistochemical (IHC) studies, tissue microarrays were incubated overnight at 4 °C with anti-Sirt6 antibodies (Abcam, Cambridge, UK). Sirt6 gene expression and clinical data were obtained from the Gene Expression Omnibus (GEO) database (http://www.ncbi.nlm.nih.gov/geo/, accessed on 18 February 2022), specifically dataset GSE29272, and The Cancer Genome Atlas (TCGA) datasets. These data were used to compare Sirt6 gene expression between normal gastric tissue and gastric cancer samples, and the association between Sirt6 expression and survival rates was evaluated.

### 2.2. Cell Lines and Chemicals

The gastric cancer cell line SNU-638 and human primary stomach epithelial cells (HPSECs) were obtained from the Korean Cell Line Bank (Seoul, Republic of Korea) and Cell Biologics, Inc. (Chicago, IL, USA). Cells were cultured in RPMI1640 medium (Thermo Fisher Scientific, Waltham, MA, USA) supplemented with 10% fetal bovine serum (GenDEPOT) and 1% penicillin/streptomycin (Thermo Fisher Scientific); HPSECs were cultured at 37 °C with 5% CO_2_ in a humidified incubator using Epithelial Cell Medium/w Kit (H6621). The chemicals used in this study were nutlin-3 (MDM2 inhibitor, Sigma-Aldrich (St. Louis, MO, USA)) and MG-132 (Calbiochem, EMD Millipore Corp., Billerica, MA, USA).

### 2.3. Western Blot Analysis

Total proteins were extracted from SNU-638 cells and HPSECs using radioimmunoprecipitation assay lysis buffer (Thermo Fisher Scientific) supplemented with a protease inhibitor cocktail (GenDEPOT). Cells were sonicated for 2 min and centrifuged at 14,000× *g* for 10 min at 4 °C to remove insoluble cell debris. Protein concentration was determined using the BCA protein assay kit (Pierce, Rockford, IL, USA). Total protein lysates (30 μg) were separated via SDS-PAGE and transferred to PVDF membranes (Millipore, Bedford, MA, USA). After being blocked with 5% BSA, the membranes were incubated with primary antibodies against Sirt1 (sc-74504, 1:500, Santa Cruz Biotechnology, Dallas, TX, USA), α-tubulin (sc-5286, 1:500, Santa Cruz Biotechnology), lamin A/C (sc-376248, 1:500, Santa Cruz Biotechnology), MDM2 (#86394S, 1:1000, Cell Signaling Technology, Inc., Beverly, MA, USA), and Sirt6 (ab191385, 1:1000, Abcam). The blots were then incubated with horseradish peroxidase-conjugated anti-rabbit IgG or anti-mouse IgG (Thermo Fisher Scientific). Signals were visualized using the Clarity Western blot ECL Substrate (Bio-Rad, Hercules, CA, USA) and imaged using the ChemiDoc Touch Imaging System (Bio-Rad).

### 2.4. Flow Cytometric DNA Analysis

Collected cells were washed twice with cold PBS, fixed with 70% ethanol for 1 h at 4 °C, treated with 1 mg/mL of RNase A (Sigma-Aldrich, St. Louis, MO, USA), and then stained with 50 μg/mL of PI (Sigma-Aldrich). Data were acquired using a Cytomics FC500 Flow Cytometer equipped with two laser sources (Beckman Coulter, Brea, CA, USA). The results were analyzed using CXP Software version 2.2 (Beckman Coulter).

### 2.5. ROS Measurement

Intracellular generation of ROS was measured using 2′,7′-dichlorodihydrofluorescene diacetate (DCF-DA; Molecular Probes, Eugene, OR, USA). Cells were stained with 5 μM of DCF-DA in a serum-free medium for 15 min and removed from the plate using trypLE-Express (Gibco, Waltham, MA, USA). Fluorescence intensity was measured using a Cytomics FC500 flow cytometer (Beckman Coulter) with an excitation wavelength of 480 nm and an emission wavelength of 525 nm. Data were analyzed using CXP Software version 2.2 (Beckman Coulter).

### 2.6. IHC Analysis of Sirt6 and Evaluation of IHC Reactions

Formalin-fixed, paraffin-embedded tissue samples from nude mice injected with SNU-638 cells were subjected to IHC. Tissue blocks were cut into 5 µm slices, which were de-paraffinized and rehydrated. The slides were incubated in 3% hydrogen peroxide for 10 min to block endogenous peroxidase activity and then heated for 20 min in 10 mM citrate buffer (pH 6.0) in a microwave oven (700 W). The sections were incubated overnight at 4 °C with anti-Sirt6 antibodies (NBP2-49668, 1:50, Novus Biological, Centennial, CO, USA).

### 2.7. ICC-Immunofluorescence (IF) for Sirt1 and Sirt6 Double Staining

HPSECs and SNU-638 cells were seeded on coverslips and treated with Ad-Lac Z or Ad-Sirt6 for 72 h. The cells were fixed with paraformaldehyde (3.7%) for 15 min at RT, washed with 1 × PBS + 0.1 M of glycine, and permeabilized with 1 × PBS + 0.2% Triton-X. The cells were then blocked with 1% BSA for 30 min at RT and stained with primary antibodies (Sirt1, sc-74504, 1:50, Santa Cruz Biotechnology; Sirt6, ab191385, 1:50, Abcam) and the corresponding secondary antibodies. Nuclei were stained with DAPI. Stained cells were analyzed using a laser-scanning confocal microscope (Olympus, Tokyo, Japan).

### 2.8. Immunoprecipitation (IP) Assay

The interaction between proteins was examined using immunoprecipitation (IP) assays. Cell lysates were first incubated with Dynabeads (10006D, Thermo Fisher Scientific Inc., Waltham, MA, USA) and Sirt6 (ab191385, 1:50, Abcam) or MDM2 (#86394, 1:100, Cell Signaling Technology) antibody for 3 h at 4 °C. The immunocomplexes were washed three times with 1× wash buffer, extracted with elution buffer, and analyzed via Western blotting using antibodies against Sirt1 and Sirt6.

### 2.9. Preparation of Adenovirus and Transduction

An adenovirus encoding *Sirt6* (Ad-Sirt6) was created using the ViraPower adenovirus expression system (Invitrogen by Thermo Fisher Scientific, USA). Briefly, cDNA encoding Sirt6 was subcloned into the pENTR vector. After sequence verification, the Sirt6 cDNA was transferred to the pAd/CMV/V5-DEST vector using the Gateway system with LR Clonase (Invitrogen). The verified clone (Ad-Sirt6) was linearized using PacI (New England Biolab), and then transfected into 293A cells using Lipofectamine 3000 (Invitrogen; Thermo Fisher Scientific, Inc.). The virus was prepared and amplified with the ViraPower adenoviral expression system (Invitrogen), and viral titers were determined using a plaque-forming assay after serial dilution. Aliquots of the viral suspension were used to infect SNU-638 cell lines. Cells were infected with adenovirus at a multiplicity of infection (MOI) of 20 and were harvested 24 to 72 h post infection for protein extraction and subsequent immunoblot analysis. A recombinant replication-defective adenovirus encoding green fluorescent protein (Ad-GFP) or β-galactosidase (Ad-LacZ) was used as a control.

### 2.10. Short Hairpin RNA (shRNA)-Mediated Silencing of Sirt6 and MDM2

For the shRNA-mediated depletion of *Sirt6* and *MDM2*, glycerol stocks of bacteria containing *Sirt6*- or *MDM2*-targeting shRNA plasmid DNA (MISSION shRNA, Sigma-Aldrich) as well as non-targeting control plasmid DNA (SHC002; Sigma-Aldrich) were purchased from Sigma-Aldrich. Lentiviral particles were used to deliver and induce the expression of shRNAs to knock down human *Sirt6* and *MDM2*, and a scrambled shRNA was used as a control. Lentiviral particles were generated via the co-transfection of a targeting set of shRNA plasmids (*Sirt6* and *MDM2*) or non-targeting control shRNA plasmid along with MISSION Lentiviral Packaging Mix (SHP001; Sigma-Aldrich) into 293FT cells (Thermo Fisher Scientific) using Lipofectamine 3000 (Life Technologies, Darmstadt, Germany). Cell culture supernatants containing lentiviral particles were collected at 24 and 48 h post-transfection, filtered, and used to infect SNU-638 cells. The efficiency of Sirt6 and MDM2 knockdown was evaluated via the Western blotting of whole-cell extracts.

### 2.11. Preparation of Nuclear and Cytosolic Extracts

Nuclear and cytoplasmic cell fractions were prepared using the NE-PER Reagent (Pierce). Briefly, cells were harvested in trypsin-EDTA (MediaTek) and spun at 500 g for 3 min. Cell pellets were resuspended in cytoplasmic extraction reagents (CERI and CERII), vortexed at high speed for 15 s, and centrifuged at 16,000× *g* for 5 min at 4 °C. Cytoplasmic protein was recovered from the supernatant. The pellet was resuspended in nuclear extraction buffer (NER) via intermittent high-speed vortexing over the course of 40 min, and the sample was centrifuged at 16,000× *g* for 10 min at 4 °C. Nuclear protein was recovered from the supernatant. The protein concentrations of the nuclear and cytoplasmic fractions were determined using a BCA assay. Cytosolic and nuclear proteins or whole-cell lysates were separated via sodium dodecyl sulfate–polyacrylamide gel electrophoresis (SDS-PAGE) in a 10% polyacrylamide gel and transferred to a nitrocellulose membrane (Millipore, Bedford, MA, USA). Membranes were incubated with primary antibodies against lamin A/C (sc-376248, 1:500, Santa Cruz Biotechnology) and α-tubulin (sc-5286, 1:500, Santa Cruz Biotechnology), followed by incubation with horseradish peroxidase-conjugated anti-rabbit IgG or anti-mouse IgG (Cell Signaling Technology, Beverly, MA, USA). Antibody binding was detected using an enhanced chemiluminescence detection reagent (Pierce). Images were acquired with the ChemiDoc Touch Imaging System (Bio-Rad).

### 2.12. Xenograft Mouse Model

All animal experiments were approved by the Institutional Animal Care and Use Committee of Gyeongsang National University (GNU-220831-M0100) and conducted according to the National Research Council Guidelines. A cell suspension (5 × 10^6^ cells/mouse) of SNU-638 was injected subcutaneously into 6-week-old male nude mice (athymic nude mice; KOATECH corporation, Harlan, Indianapolis, IN, USA). Nine days after the inoculation of the cells, animals with xenograft tumors measuring 0.6–0.7 cm in diameter were subjected to intratumoral injections of Ad-GFP or Ad-Sirt6 (5 × 10^9^ PFU). Tumor diameters were measured with digital calipers on days 5, 10, 15, and 20, and tumor volume was determined using the modified ellipsoidal formula (tumor volume = 1/2[length × width^2^]).

### 2.13. Statistics

All statistical analyses were performed using SPSS software, version 20.0 (IBM, Armonk, NY, USA), and GraphPad Prism (version 8.0; GraphPad Software, San Diego, CA, USA). Chi-squared analysis was used to assess the relationship between Sirt6 expression and clinicopathological parameters. Student’s *t*-test was used to analyze the differences between groups and among groups, respectively. In all the tests, a *p*-value < 0.05 was used as the cut-off for statistical significance.

## 3. Results

### 3.1. Sirt6 Is Downregulated in Gastric Cancer, and Its Higher Expression Levels Are Linked to Improved Patient Survival

Sirt6 expression was significantly lower in gastric cancer tissues than in normal tissues (Figure 1A). Sirt6 expression levels varied among gastric cancer cells, and high expression levels were associated with higher overall survival rates (Figure 1B,C).

### 3.2. Manipulation of Sirt6 Levels Affects Cell Death Rates and Tumor Volume in Gastric Cancer Models

Sirt6 expression levels were lower in SNU-638 cells (gastric cancer cell line) than in the control cells (HPSECs) (Figure 2A). Sirt6 overexpression provoked using an adenovirus induced gastric cancer cell death, as indicated by the percentage of Sub G1 cells, whereas no effect was observed in HPSECs (Figure 2B). The knockdown of Sirt6 via shSirt6 treatment decreased gastric cancer cell death (Figure 2C). In a SNU-638 xenograft mouse model, Sirt6 overexpression significantly decreased gastric cancer tumor volume, suggesting that Sirt6 exerts anti-tumorigenic effects (Figure 2D,E).

### 3.3. Modulation of Sirt6 Influences ROS Production and Apoptotic Responses in Gastric Cancer Cells

Because oxidative stress is a key factor regulating the tumor-promoting effect of sirtuins, we measured ROS levels in gastric cancer cells. Sirt6 overexpression induced by Ad-Sirt6 increased ROS accumulation, whereas Sirt6 knockdown via shSirt6 decreased ROS production in gastric cancer cells (Figure 3A,B). The ROS scavenger NAC significantly reduced Sirt6-mediated gastric cancer cell apoptosis (Figure 3C).

### 3.4. Changes in Sirt6 Expression Alter Sirt1 Levels, Affecting Gastric Cancer Cell Apoptosis

Ad-Sirt6-induced Sirt6 overexpression downregulated Sirt1 in gastric cancer cells (Figure 4A), whereas shSirt6-mediated Sirt6 silencing upregulated Sirt1 expression (Figure 4B). Sirt6 overexpression in gastric cancer cells was predominantly detected in the nucleus and not in the cytoplasm, as indicated by Western blot and immunofluorescence analyses (Figure 4C,D).

### 3.5. Proteasome Inhibition and MDM2 Modulation Significantly Affect Sirt1 and Sirt6 Levels

Oxidatively damaged proteins are known to be degraded via the proteasome system, and treatment with MG-132, a proteasome inhibitor, upregulated Sirt1 (Figure 5A). Suppressed gastric cancer cell apoptosis was associated with Sir6 (Figure 5B). The knockdown of MDM2 via treatment with sh-MDM2 upregulated Sirt1 leading to reduced apoptotic activity (Figure 5C,D). Consistently, the inhibition of the p53-MDM2 interaction induced by Nutlin-3 suppressed Sirt1- and Sirt6-mediated apoptotic activity (Figure 5E,F). Additionally, Sirt1 downregulation and Sirt6 upregulation were detected in the presence of MDM2 in gastric cancer cells (Figure 5G).

A schematic diagram describing Sirt6- and Sirt1-mediated gastric cancer cell death is shown in Figure 6. MDM2-modulated Sirt6 upregulation and Sirt1 downregulation promoted gastric cancer cell death via increasing ROS accumulation (Figure 6).

## 4. Discussion

Gastric cancer is among the leading causes of cancer-related death, although the underlying pathogenetic mechanism remains unclear [41]. *Helicobacter pylori* infection is associated with an increased risk of developing gastric cancer; however, *H. pylori* infection is not present in all gastric cancer cases, and other factors such as inflammatory conditions and alterations in hormone levels have been associated with gastric cancer development [42,43]. Among several oncogenic factors, members of the sirtuin family play a crucial role in the immune response to gastric cancer [10]. This study is the first to examine the role of Sirt1 and Sirt6 in the regulation of gastric cancer cell death.

Sirt6 exhibits anti-cancer effects in many malignancies; however, whether Sirt1 acts as a pro-tumorigenic or anti-tumorigenic factor remains unclear. The cancer-related effects of Sirt1 or Sirt6 have also been reported in relation to gastric cancer, and the corresponding studies are listed in Table 1. The pathways and molecules involved in both the tumorigenic and anti-tumorigenic effects of Sirt1 include YAP, FOXO, p53, and miRNAs, whereas the JAK2/STAT3 pathway and ferroptosis are involved in the anti-tumorigenic effect of Sirt6 in gastric cancer. This study focused on ROS-mediated cancer cell death because ROS are associated with a variety of cellular processes involving both Sirt1 and Sirt6 [44,45].

In corroborating the regulatory cascade leading to ROS-mediated cell death, we identified MDM2 as a critical upstream modulator of Sirt1 activity. This selection is grounded in research demonstrating that MDM2, known for its regulatory effect on p53, also exerts influence over Sirt1. For instance, Park et al. (2016) have elucidated the anti-cancer effects of the SIRT inhibitor MHY2256 on MCF-7 breast cancer cells through the modulation of the MDM2-p53 interaction, which is intimately connected to Sirt1 regulation [46]. Furthermore, a study by Grabowska et al. (2017) provided insights into how Sirt1 can inhibit Suv39h1 methyltransferase degradation by impeding MDM2-mediated polyubiquitination [47]. These studies provide a foundational basis for our rationale in targeting the MDM2-Sirt1 axis in the context of gastric cancer.

Sirt6- and Sirt1-mediated gastric cancer cell death via ROS regulation was demonstrated previously in many types of cancer but not in gastric cancer [6]. We showed that Sirt6 upregulates MDM2, which suppresses Sirt1 expression and induces ROS production, thereby promoting gastric cancer cell death. Sirt1 suppression itself does not lead to cancer cell death, requiring Sirt6- and MDM2-mediated downregulation to induce an anti-cancer effect. The mechanisms and molecular pathways involved in Sirt1 suppression need to be examined further to develop future Sirt-based anti-cancer therapeutics.

## 5. Conclusions

In summary, we found that elevated Sirt6 expression leads to Sirt1 suppression via MDM2, inducing gastric cancer cell death. The activity of the three proteins, Sirt6, MDM2, and Sirt1, results in increased ROS production, which is responsible for an anti-tumorigenic effect, suggesting that these three molecules are potential targets for the treatment of gastric cancer.

## Figures and Tables

**Figure 1 cancers-16-00387-f001:**
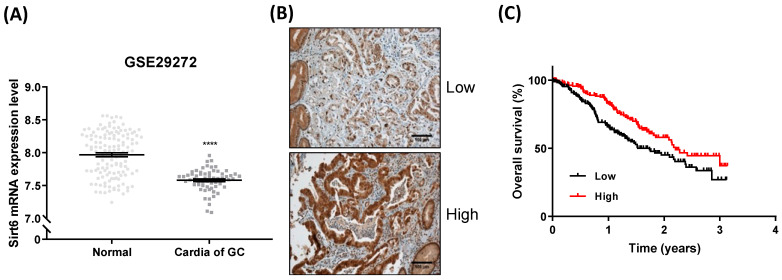
Sirt6 expression in gastric cancer. (**A**) Analysis of GSE29272 revealed that Sirt6 mRNA expression levels were lower in gastric cancer cells than in normal cells (**** *p* < 0.0001, *t*-test). (**B**) Representative images of low and high Sirt6 expression in gastric cancer tissues. (**C**) Overall survival analysis using a Kaplan–Meier plot based on The Cancer Genome Atlas (TCGA) dataset showed that high expression of Sirt6 was associated with higher overall survival among gastric cancer patients.

**Figure 2 cancers-16-00387-f002:**
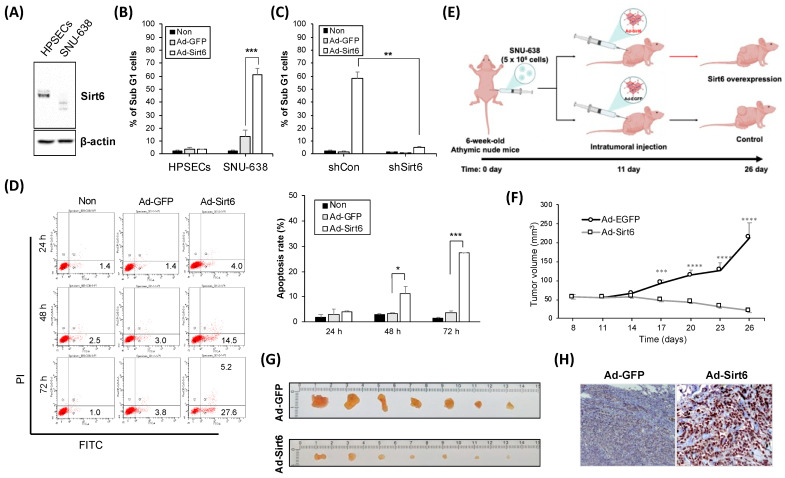
Sirt6 expression is associated with gastric cancer cell death and a higher survival rate. (**A**) Western blot analysis showing that Sirt6 was downregulated in gastric cancer cells (SNU-638) compared with human primary stomach epithelial cells (HPSECs). (**B**) Quantitative analysis of cell death indicating that adenovirus-mediated overexpression of Sirt6 significantly induced cell death in gastric cancer cells (*** *p* < 0.001). (**C**) Bar graph depicting Sirt6 knockdown decreasing gastric cancer cell death (** *p* < 0.01). (**D**) Dead cells were analyzed by flow cytometry after double staining with Annexin V-FITC and PI. The amounts of apoptosis were determined as percentages of Annexin V+ and/or PI+ cells. (**E**) A schematic representation of the xenograft model timeline detailing the process from cell injection to treatment administration and subsequent measurement of tumor volumes. Figure created with BioRender.com. (**F**) Graph and (**G**,**H**) tumor tissues showing that Sirt6 overexpression led to a significant reduction in tumor volume in the gastric cancer xenograft model, with measurements taken at specified time points post treatment (**** *p* < 0.0001,*** *p* < 0.001, ** *p* < 0.01).

**Figure 3 cancers-16-00387-f003:**
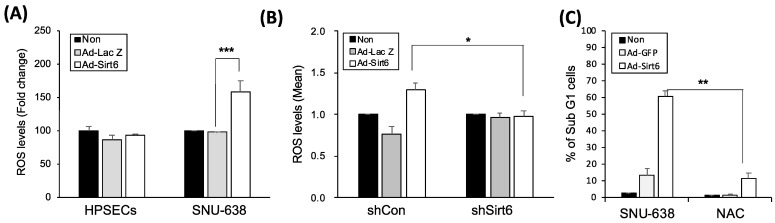
Sirt6 induces ROS-mediated cell death in gastric cancer. (**A**) Sirt6 overexpression increased ROS accumulation in gastric cancer cells, whereas it had no effect on ROS levels in HPSECs (*** *p* < 0.001). (**B**) Sirt6 knockdown via shSirt6 decreased ROS levels (* *p* < 0.05). (**C**) The ROS scavenger NAC inhibited gastric cancer cell death (** *p* < 0.01).

**Figure 4 cancers-16-00387-f004:**
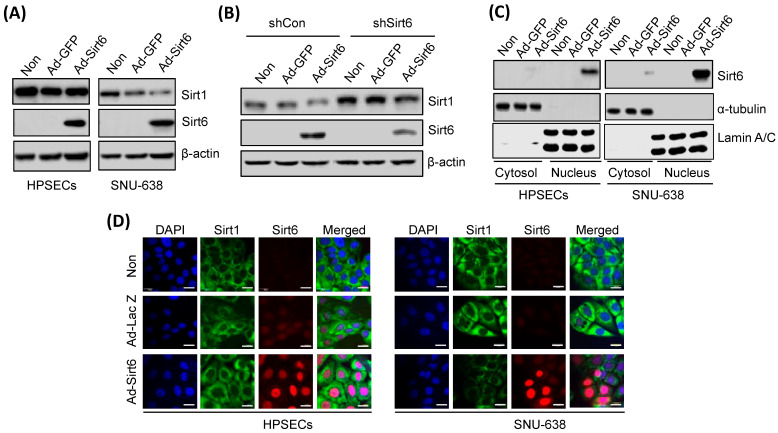
Sirt6 overexpression downregulates Sirt1 in gastric cancer. (**A**) Sirt6 overexpression induced via Ad-Sirt6 transfection downregulated Sirt1 expression in gastric cancer, whereas it had no effect on HPSECs. (**B**) Sirt6 knockdown induced via shSirt6 upregulated Sirt1. (**C**) Sirt6 expression was predominantly detected in the nucleus rather than in the cytosol. (**D**) Fluorescence images showed that treatment with Ad-Sirt6 upregulated Sirt6 and downregulated Sirt1. The scale bar represents 100 μm.

**Figure 5 cancers-16-00387-f005:**
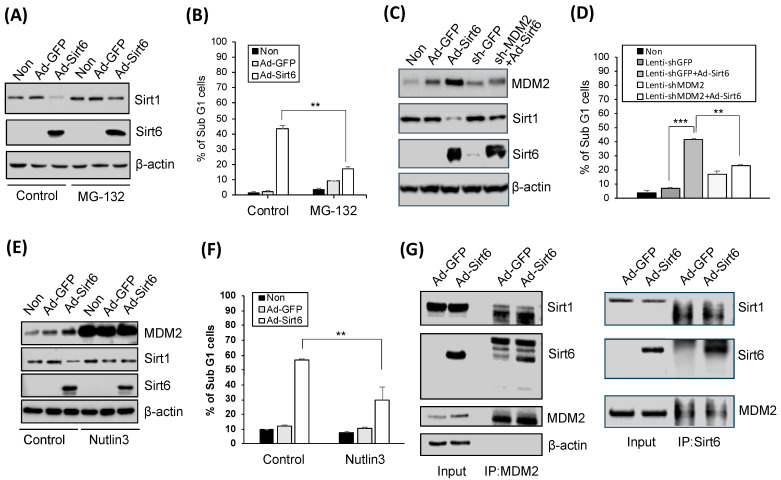
Sirt6-induced gastric cancer cell death is mediated by Sirt1 suppression. (**A**) Treatment with 2.5 µM MG-132 (a 26S proteasome inhibitor) for 8 h upregulated Sirt1. (**B**) Apoptosis, as indicated by the percentage of Sub G1 cells, was significantly decreased via the MG-132 treatment (** *p* < 0.01). (**C**) Sirt6 overexpression induced MDM2 and downregulated Sirt1 expression, whereas MDM2 knockdown upregulated Sirt1. (**D**) Gastric cancer cell death was induced by Sirt6 overexpression and reduced by MDM2 knockdown (*** *p* < 0.001, ** *p* < 0.01). (**E**) Nutlin-3, an MDM2 antagonist, upregulated Sirt1 but did not affect Sirt6 expression levels. (**F**) Nutlin-3 inhibited gastric cancer cell death (** *p* < 0.01). (**G**) Sirt1 was downregulated while Sirt6 was upregulated in the presence of MDM2 in gastric cancer cells.

**Figure 6 cancers-16-00387-f006:**
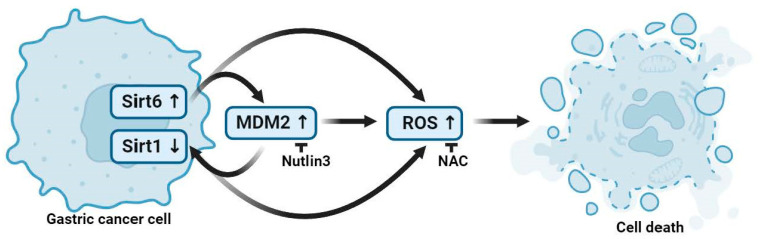
A schematic illustration of gastric cancer cell death mediated by Sirt6 and Sirt1. MDM2 expression in association with Sirt6 upregulation and Sirt1 downregulation leads to ROS accumulation, and the increased ROS expression level promotes gastric cancer cell death. Figure created with BioRender.com.

**Table 1 cancers-16-00387-t001:** Dual roles of Sirt1 and Sirt6: modulating tumorigenesis and anti-tumorigenic pathways.

Name	Function	Mechanism	References
**Sirt1**	**Pro-tumorigenic**	Yap signaling pathway	[11,12]
		Modulation of angiogenesis via FOXO1	[13]
		Deacetylation of histone substrates, transcription factors, and cofactors	[14]
		FOXO1 and YAP signaling pathway modulated by USP22	[15]
		Regulating autophagy through the deacetylation of ATGs	[16]
		Targeting p53 to suppress ferroptosis	[17]
		Mediated by candidate oncogene circNOP10	[18]
		FoxO1-Rab7–autophagy axis	[19]
		Deacetylation of Beclin-1 and other autophagy mediators	[20]
		Inhibiting PI3K/AKT pathway	[21]
		Sirt1 downregulation by MiR-204	[22]
		By the results of DBC1, H4K16Ac, and H3K9Ac	[23]
		Sirt1 transactivation by ATF4	[24]
		Nampt/sirt1/c-myc positive feedback loop	[25]
	**Anti-tumorigenic**	Sirt1 targeted by micro-RNAs (miR-543, miR-132-3p/miR-212-3p, miRNA-12129, miR-1301-3p, Has-miR-34a-5p, miR-132, miR-543, miR-183)	[15,26,27,28,29,30,31,32]
		Regulating ARHGAP5 expression	[33]
		Initiating an AMPK/FOXO3 positive feedback loop	[34]
		Via STAT3/MMP-12 signaling	[35]
		G1-phase arrest via NF-κB/Cyclin D1 signaling	[19]
		Repression of activation of STAT3 and NF-κB proteins via deacetylation	[36]
		Inducing G1 phase arrest and senescence by resveratrol	[37]
**Sirt6**	**Anti-tumorigenic**	Inhibition of JAK2/STAT3 pathway	[38]
		Promoting ferroptosis	[39]

## Data Availability

Data are contained within the article and Appendix A.

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
