# Peer review of "Sirt6-Mediated Cell Death Associated with Sirt1 Suppression in Gastric Cancer"

_cancers, 2024, doi:10.3390/cancers16020387_

Round 1

Reviewer 1 Report

Comments and Suggestions for Authors

The study conducted by Ji Hyun Seo et al. on the molecular mechanisms of Sirt6-mediated cell death and its suppression of Sirt1 in gastric cancer is conceptually interesting. However, the manuscript is currently undermined by substantial linguistic deficiencies that obscure its scientific merits. The results are presented in a manner lacking in clarity and scientific rationale, making comprehension challenging. It is imperative that the authors undertake rigorous language editing to meet academic standards and to ensure the logical and clear conveyance of their findings. Improving these critical aspects is essential for the manuscript to be considered seriously in the scientific community

Here are a few comments:

Major comments:

1.     It’s hard to believe that the authors think “allow unchecked cell division” is an anti-tumorigenic effect; “protect DNA from unfavorable environmental conditions such as oxidative stress, maintain genomic stability” is a pro-tumorigenic effect. This is a misinterpretation of the references and an act of confusing right and wrong.

2.     The overall resolution of all the figures is too low, they need to be extensively revised, especially Figure 2 E, as it’s hard to tell what things were there. Some of the fronts are very difficult to distinguish, such as Figure 1 A, Figure 5 A, C, E, G et al.

3.     The meaning of Figure 1 B is unclear. The author should provide information including 1) what experiment it is? 2) how many samples were used? 3) how does this data support “Sirt6 expression levels varied among gastric cancer cells, and high expression levels were associated with higher overall survival rates”? 

4.     Line 52, I didn’t see “table 1”.

5.     The authors should provide the western blot data to confirm the successful overexpression or knockdown of Sirt6 in these experiments: Figure 2B, 2C, 2E, 3A, 3B.

6.     Figure 2A, the author should describe the reason why they chose to use the SNU-638 cell line for this study.

7.     Figure 2B, The author should explain Why the SNU-638 Ad-GFP group has as high as 15% sub-G1? 

8.     The author should explain Why the Sirt6 band exhibits at around 40kDa in the HPSECs cell line but 55kDa in the SNU-638 cell line in Figure 4A? 

9.     Why Figure 4B Sirt6 band size is different from Figure 4A?

10.  The author should explain why over-expressed Ad-GFP significantly reduces Sirt1 expression in the SN-638 cell line in Figure 4A.

Minor comments:

11.  The introduction part is too general, such as “The anti-tumorigenic activity of Sirt6 is mediated by various mechanisms…” “Sirt1 has a dual function as a pro-tumorigenic and anti-tumorigenic factor according to the environment and cell type” et al. This part needs to be completely revised. The authors should describe clearly what is known and what is unknown about Sirt6 and Sirt1. Why did they select gastric cancer for the study? What’s the advance in comparison with their previous research published in experimental & molecular medicine? What’s the scientific question they would like to solve?

12.  The author should provide detailed information about the antibodies they used in this study, including their catalog number, lot number, and dilution ratio in different assays such as IHC and Western blot et al.

Comments on the Quality of English Language

A lot of spelling and gramma mistake, needs extensive language editing.

Reviewer 2 Report

Comments and Suggestions for Authors

1. A subtitle in the section result would be great to be consistent with each figure.

2. Antibobodies dilution and catalog numbers are necessary to be in the context.

3. In general, I would suggest using a sirt6 pharmacological inhibitor to perform rescue experiments since it is practically easier to perform and causes less confusion.

4. It looks like sirt6 is a tumor suppressor in gastric cancer cell line based on your data. In that case, have you tried to knock out sirt6 to see if the tumor becomes more aggressive? 

Figure 1: Since IHC was performed on human samples, the expression of sirt6 in WB should also be addressed.

Figure 2:

1. What do you mean by "downregulated" in Figure 2 panel A? Does it mean post-translational modification in SNU638? Please clarify.

2. Please clarify the viral volume and time for the sirt6 transduction.  

3. Flow charts of the apoptosis should be also presented to indicate the subG1 population of each sample. Also Annexin V staining could be applied to check cell apoptosis.  

4. An animal model timeline scheme is necessary to demonstrate the xenograft study.

Figure 4

Merge images and scale bar of Immunofluorescence are necessary, and it looks like the images of the SNU638 cell line transduced with Sirt6 are not consistent. 

Figure 5

1. The dose and time of MG-132 treatment are not indicated.

2. How did you screen out MDM2 as the upstream of sirt1? By checking the literature (please indicate in citation) or bioinformatic analysis (please show the data)?

3. In panel C, Why there is a weak band in shGFP probed by sirt6? Also a knockdown expression of MDM2 should be addressed to check the efficiency. 

Panel G, IgG should be added as a negative control. Since sirt6 is overexpressed in this cell line, it would be practically available to use anti-sirt6 to pull down.

Comments on the Quality of English Language

This manuscript would benefit from a thorough review by an individual proficient in technical English. Special attention should be directed towards improving grammar, spelling, and sentence structure, ensuring the objectives and findings of the study are clearly communicated to readers. The words in any part of this work need to be defined scientifically, here are some examples:

"Oxidatively damaged proteins are known to be degraded via the proteasome system 234 and treatment with MG-132, proteasome inhibitor, upregulated Sirt1" "Because oxidative stress is a key factor regulating the tumor-promoting effect of sirtuins, we measured ROS levels in gastric cancer cells. "

Reviewer 3 Report

Comments and Suggestions for Authors

The study investigates the role of Sirt6 and Sirt1 in gastric cancer tumorigenesis. Gastric cancer is strongly associated with H. pylori infection and is one of the leading causes of cancer-related death. The sirtuin family of proteins has been found to be involved in the development of gastric cancer, with both pro-tumorigenic and anti-tumorigenic effects. The researchers investigated the effect of Sirt6 upregulation on gastric cancer cells using both in vitro and in vivo experiments. They measured the overall survival rate of gastric cancer patients with or without Sirt6 expression and found that Sirt6 overexpression resulted in a reduction in tumor volume in vivo. They further analyzed the regulation of Sirt6 and Sirt1 and their downstream effects on MDM2 and reactive oxygen species (ROS) production in gastric cancer cells. They used western blot and immunofluorescence staining to measure protein expression levels and flow cytometric analysis to determine cell death rates in response to Sirt6 and Sirt1 modulation. Their results showed that Sirt6 overexpression led to the suppression of Sirt1 in gastric cancer cells, resulting in increased ROS and MDM2 expression levels and ultimately leading to increased cell death. These findings suggest that Sirt6 and Sirt1 may be potential therapeutic targets in gastric cancer. In conclusion, this study provides new insights into the role of Sirt6 and Sirt1 in gastric cancer tumorigenesis and highlights their potential as therapeutic targets. Further research is needed to fully understand the mechanisms underlying these effects and to develop effective treatments for gastric cancer. This an interesting and informative manuscript. But I have several following concerns: 

1. Abbreviations should be defined when they first appear in the text. Such as "MDM2", "NAD+",...

2. Linnaean nomenclature"H. pylori ", the species name should be written in full when they first appear.

3. Please provide the ethical approval number of the animal experiment in the article

4. For Western blot Figures, it is better for the author to mark the approximate position of the protein with Marker and perform statistical analysis on the Figures.

5. For Figure 4, please add scale bar for Fig 4D. 

6. If Figures are a copy of the reported Figures or made with software, please apply for copyright or indicate the quotation.

7. The nucleic acid sequences (including gene names, regulatory sequences, and primer names) should be in italics.

8. Please unify the format of references in the article, including the author's name, the case of words in the title of the article, the writing of the name of the journal, and the page number.

Comments on the Quality of English Language

Moderate editing of English language required.

Round 2

Reviewer 1 Report

Comments and Suggestions for Authors

The manuscript has been greatly improved. I don't have more comment.

Author Response

We appreciate your valuable comments.

Reviewer 2 Report

Comments and Suggestions for Authors

Great improvement after the revision. Some minor issues need to be revised as follow:

1. IF Merge images should include DAPI. 

2.

à We initially focused on ROS-related anti-cancer effect of Sirt1, and MDM2 was screened out based on checking several literatures. These papers are not added to the reference list since we did not directly mention the reason why MDM2 was screened out as the upstream of Sirt1 in the manuscript. For example, Park EY et al. reported that anticancer effect of a new Sirt inhibitor, MHY2256, against human breast cancer cells was regulated by MDM2-p53 binding (Int J Bilo Sci, 2016), and Grabowska W et al. showed that Sirt1 inhibited Suv39h1 methyltransferase degradation by inhibiting polyubiquitination of this methyltransferase by MDM2 (Biogerontology, 2017).

One or two sentences should be added into the context to introduce MDM2 as the upstream of sirt1 along with those citations you mentioned (should be listed in the reference), which may greatly help readers understand the work of 3.5. 

Reviewer 3 Report

Comments and Suggestions for Authors

The authors have addressed all my concerns. I recommend accepting it in current form. 
